# A Glimpse at the Size of the Fetal Liver—Is It Connected with the Evolution of Gestational Diabetes?

**DOI:** 10.3390/ijms22157866

**Published:** 2021-07-23

**Authors:** Matei-Alexandru Cozma, Mihnea-Alexandru Găman, Elena-Codruța Dobrică, Steluța Constanța Boroghină, Mihaela Adela Iancu, Sanda Maria Crețoiu, Anca Angela Simionescu

**Affiliations:** 1Faculty of Medicine, “Carol Davila” University of Medicine and Pharmacy, 050474 Bucharest, Romania; matei.cozma@gmail.com; 2Department of Gastroenterology, Colentina Clinical Hospital, 20125 Bucharest, Romania; 3Department of Hematology, Center of Hematology and Bone Marrow Transplantation, Fundeni Clinical Institute, 022328 Bucharest, Romania; 4Department of Pathophysiology, University of Medicine and Pharmacy of Craiova, 200349 Craiova, Romania; codrutadobrica@gmail.com; 5Department of Dermatology, “Elias” University Emergency Hospital, 011461 Bucharest, Romania; 6Department of Complementary Sciences, History of Medicine and Medical Culture, “Carol Davila” University of Medicine and Pharmacy, 050474 Bucharest, Romania; steluta.boroghina@umfcd.ro; 7Department of Family Medicine, “Carol Davila” University of Medicine and Pharmacy, 050474 Bucharest, Romania; adelaiancu@yahoo.com; 8Department of Cell and Molecular Biology and Histology, “Carol Davila” University of Medicine and Pharmacy, 050474 Bucharest, Romania; 9Department of Obstetrics and Gynecology, Filantropia Clinical Hospital, “Carol Davila” University of Medicine and Pharmacy, 050474 Bucharest, Romania; anca.simionescu@umfcd.ro

**Keywords:** fetal liver, gestational diabetes, dietary patterns, obstetrical ultrasound, pregnancy, pregnancy complications

## Abstract

Gestational diabetes mellitus (GDM) is defined as an impairment of glucose tolerance, manifested by hyperglycemia, which occurs at any stage of pregnancy. GDM is more common in the third trimester of pregnancy and usually disappears after birth. It was hypothesized that the glycemic status of the mother can modulate liver development and growth early during the pregnancy. The simplest modality to monitor the evolution of GDM employs noninvasive techniques. In this category, routinely obstetrical ultrasound (OUS) examinations (simple or 2D/3D) can be employed for specific fetal measurements, such as fetal liver length (FLL) or volume (FLV). FLL and FLV may emerge as possible predictors of GDM as they positively relate to the maternal glycated hemoglobin (HbA1c) levels and to the results of the oral glucose tolerance test. The aim of this review is to offer insight into the relationship between GDM and fetal nutritional status. Risk factors for GDM and the short- and long-term outcomes of GDM pregnancies are also discussed, as well as the significance of different dietary patterns. Moreover, the review aims to fill one gap in the literature, investigating whether fetal liver growth can be used as a predictor of GDM evolution. To conclude, although studies pointed out a connection between fetal indices and GDM as useful tools in the early detection of GDM (before 23 weeks of gestation), additional research is needed to properly manage GDM and offspring health.

## 1. Introduction 

Gestational diabetes mellitus (GDM) is a relatively common pregnancy pathological condition that was recently defined by the American Diabetes Association (ADA) as hyperglycemia, with no obvious cause, first appearing or discovered during the pregnancy’s second or third trimester [1,2,3,4,5]. It was suggested to include in this definition the preexisting, nonidentified cases of type 2 diabetes mellitus (“overt diabetes”) and type 1 diabetes mellitus, but these are detected very early after the onset of the pregnancy [6,7]. However, GDM develops later during the pregnancy and is usually detected between week 24 and week 28 of gestation [8].

Maternal obesity has emerged as a public health threat and a worrying elevation in its prevalence has been registered worldwide. It is well known that obesity contributes to the development of GDM and of cardiovascular disease (CVD) both in pregnant and nonpregnant females [9,10]. During the last three decades, the number of diagnosed cases of GDM has increased by 10–100% [11,12,13]. The worldwide prevalence of GDM is estimated at 7% of all pregnancies, but it is difficult to correctly estimate it since there are no precise screening criteria for GDM, as comprehensively reviewed by Caissutti et al. (2017) [14]. The prevalence of GDM is difficult to establish since there is still no general international consensus on the methods used for screening. According to the ADA, prenatal testing of the population with risk factors, testing of all pregnant women in the second trimester, and testing of women with GDM at one to three months postpartum is recommended. Although the oral glucose tolerance test is fast, it is associated with a high degree of discomfort in comparison with the obstetrical ultrasound (OUS), which can be performed at the same time with routine ultrasound scans [15,16,17].

A relatively recent meta-analysis of 40 studies involving a total of 177,063 subjects displayed that the prevalence of GDM in Europe is 5.4% [18]. The incidence of GDM differs depending on the diagnostic guidelines and the cutoff values employed, respectively, and is currently estimated at 14% of all pregnancies worldwide. Thus, GDM affects around 18 million pregnancies annually [2]. The exact incidence is difficult to establish as the limits of the range vary significantly, i.e., from 2% to 37% [19,20,21]. The International Diabetes Federation estimated that 21.3 million live births worldwide are affected by some type of hyperglycemia in pregnancy, out of which 83% are due to GDM. Meanwhile, one in six pregnancies is affected by GDM [22]. The prevalence of hyperglycemia in pregnancy varies between different geographical areas from 10.4% in North America and the Caribbean Region to 25.0% in Southeast Asia [13]. Ethnicity influences the risk of GDM, with Asian women having an increased risk versus other ethnic groups [23]. The incidence of GDM has been steadily increasing, mainly due to the increase in the age of the pregnancy and, most importantly overall, the weight of women [24]. Risk factors for the development of GDM include a family history of maternal overweight/obesity or diabetes, age of the mother >35 years, smoking, use of a Western diet, micronutrient deficiencies, multiparity, a history of dysglycemia, personal history of GDM or previous pregnancy with a macrosomic fetus (newborn above 4 kg). (Figure 1) [25,26,27,28,29,30]. A personal history of hyperglycemia or the presence of GDM in a previous pregnancy increases the risk of GDM recurrence in subsequent pregnancies [29]. Women in whom the presence of hirsutism and/or hyperandrogenism without a diagnosed polycystic ovary syndrome or other clinical conditions associated with insulin resistance (e.g., obesity, acanthosis nigricans) is noted seem to display elevated odds of GDM versus females without polycystic ovary syndrome or the aforementioned conditions [28,31,32]. In addition, women diagnosed with hypertension have an increased risk of GDM versus normotensive females [33]. The use of several drugs, e.g., antidepressants, antipsychotics, beta-adrenergics, or corticosteroids, has been linked with an increased risk of GDM as well [34,35,36]. In addition, some other factors that can be incriminated in the development of GDM are macrosomia (exaggerated somatic development of the fetus) during the current pregnancy or 2 or more episodes of glycosuria during the second or third trimester of gestation [2,5].

The pathophysiology of GDM is not fully understood, but the latest studies point out that in GDM there is an elevated insulin requirement and a progressive peripheral resistance to the action of this hormone, most often present but not expressed before the pregnancy. Various circulating cytokines, e.g., interleukin 6 (IL−6) or tumor necrosis factor-alpha (TNF-alpha), seem to be culprits in exacerbating insulin resistance in pregnancy [3].

The standard diagnostic method for GDM is the oral glucose tolerance test (OGTT) (75 g). Thus, all pregnant women should be subjected to this test at 24–28 weeks of gestation, although a recent study showed that GDM diagnosed at 24–28 weeks of gestation has already impacted the fetus but only in older and obese women, suggesting the necessity of screening before this period [8]. The cutoff values for the OGTT are blood glucose of 92, 180, and 153 mg/dL, before administration of 75 g of oral glucose and one and two hours after, respectively [2,4,5,37]. Although this method is superior to the measurement of fasting blood glucose or glycated hemoglobin (HbA1c), it also has some disadvantages, such as high cost and duration, low reproducibility, and discomfort caused to the patient by the high amount of carbohydrates that has to be ingested in a very short period of time [5,37].

Although the above tests have an undeniable diagnostic value, new noninvasive diagnostic methods, e.g., ultrasonography, have emerged as useful tools in identifying patients at risk of developing GDM. Thus, Perovic et al. (2012) highlighted the advantages of an ultrasound GDM screening score as a predictor of GDM development by screening pregnant females who were at least in the week 24 of gestation and who harbored risk factors for GDM. The ultrasound GDM screening score proposed by the aforementioned researchers exhibited a specificity and a sensitivity of over 89% and took into consideration several relevant parameters, i.e., subcutaneous fetal adipose tissue, the immature appearance of the placenta, and the placental thickness [38]. Moreover, another study by Gojnic et al. (2012) revealed that subcutaneous fetal adipose tissue exhibited the best specificity and sensitivity in predicting GDM if the evaluation was performed in week 32 of gestation [39]. Furthermore, prior to week 24 of gestation, other ultrasound parameters, i.e., head circumference below the 10th percentile and femur length below the 10th percentile have been linked with a 13–17% elevated risk of GDM [40].

Behavioral changes such as diet modification and increased physical activity are often used in the first-line treatment of GDM [41,42]. If glucose levels are not adequately controlled via these strategies, pharmacological agents, e.g., metformin, glibenclamide (glyburide), and/or insulin, can be effectively employed [43]. A personalized approach to pharmacological treatment is recommended [44]. The use of metformin in patients with GDM should not neglect the negative impact of this drug on the offspring’s development or the high risk of maternal hypoglycemia. Therefore, metformin should be prescribed with precaution considering that it is not associated with an improvement in insulin sensitivity and there is a need to clarify whether it can cause any harm to the offspring in the long term [44,45,46,47]. This is particularly worrying given that the offspring of females with untreated GDM are prone to develop obesity, diabetes, and CVD, an important factor that will contribute towards the burden of diabetes reaching 439 million patients worldwide (approximately 70% of adults aged 54–60) by 2030 [2,48]. Insulin has low efficacy due to the degree of peripheral resistance to its action, and the long-term effects of oral antidiabetics, such as metformin or glyburide, both on the mother and the fetus, are not yet fully understood, as they also have a high number of side effects, such as maternal weight gain and neonatal hypoglycemia [2,49].

## 2. Dietary Factors and Dietary Patterns Associated with the Development of GDM

The development of GDM can occur under the influence of several factors, as previously described. However, certain dietary patterns before and during the pregnancy might play roles in its development as well. Although the attention of many researchers has been focused on several macro- and (or) micronutrients, we believe a broader approach would be more suitable for women at risk of GDM. Thus, in the future, special attention should be given to dietary patterns. One discusses here some of the most recent conclusions of the studies in the field.

A study by Shin et al. (2015) of 253 pregnant women included in the National Health and Nutrition Examination Survey (NHANES) concluded that there is a direct link between “a high consumption of refined grains, fat, added sugars and low intake of fruits and vegetables” and the risk to develop GDM during pregnancy. However, it also underlines the need for future prospective and cohort studies that should be correlated with the degree of physical activity [50]. Sedaghat et al. (2017) evaluated 388 pregnant women and analyzed the role of maternal dietary patterns in the progress of GDM. The Western dietary pattern (rich in “sweets, jams, mayonnaise, soft drinks, salty snacks, solid fat, high-fat dairy products, potatoes, organ meat, eggs, red meat, processed foods, tea, and coffee”) was linked with an elevated GDM risk [51]. In their systematic review and meta-analysis, Mijatovic-Vukas et al. (2018) pointed out that diets resembling the MedDiet/DASH (Mediterranean diet/dietary approaches to stop hypertension) diets, if associated with physical exercise before and during the first semester of pregnancy, are linked with a reduced risk of GDM [52]. Hu et al. (2019) conducted a prospective cohort study on the Chinese population and demonstrated that a “traditional pattern” (rich in vegetables, fruits, and rice) was protective against GDM development versus a whole grain–seafood dietary pattern [53]. In their review, Moses et al. (2009) concluded that in subjects who have a high-fiber, low-glycemic index diet and practice physical exercise daily, one serving of sugar-sweetened beverages on a daily basis is unlikely to be associated with a higher GDM prevalence [54]. Based on data from a prospective dynamic cohort study, elevated consumption of fast food (pizza, sausages, and hamburgers) has emerged as an independent risk factor for GDM in 3048 pregnant Spanish females [55]. A longitudinal cohort study on twin pregnancies analyzed four dietary patterns and concluded that none of the studied dietary patterns was associated with the risk of GDM in twin pregnancies. A significant increase in the risk of GDM was observed only among normal-weight women prior to pregnancy who had a sweet-based diet pattern [56]. Prospective research coming from central China revealed that a diet rich in proteins and low in carbohydrates during midpregnancy was associated with a higher risk of GDM, although the underlying mechanisms remain unexplained. The association of the fish–meat–eggs pattern with GDM risk was stronger in females suffering from overweight or who had a family history positive for diabetes [57]. In their research, Zhou et al. (2018) indicated that a diet rich in rice–wheat–fruits pattern was linked with a lower GDM risk, but this protective effect was not discovered in females with an elevated prepregnancy body mass index (BMI) or in those who had a positive family history for diabetes [57].

There also seems to be a connection between the diet and the microbiome of pregnant females suffering from GDM, with the microbiome viewed as another culprit in enhancing insulin resistance and a proinflammatory state in GDM [58]. Moreover, the dysfunctional microbiome of the mother can be passed on to the offspring, which can later develop allergies, respiratory diseases (e.g., asthma), cardiometabolic disorders (obesity, diabetes, CVD), or neurological diseases in a direct relationship with the inherited dysbiosis [59].

For example, in terms of phylum, Actinobacteria seem to be more represented in pregnant women with GDM. In addition, in terms of the genus, *Desulfovibrio*, *Rothia*, and *Collinsella* can be detected in higher amounts in GDM pregnancies [60]. Another paper points out that there is an abundance of *Prevotella*, *Parabacteroides distasonis*, and *Ruminococcaceae* in pregnant females suffering from GDM [61]. In murine models, high consumption of fructose during the pregnancy was linked with a decrease in *Bacteroides* and *Lactobacillus*, which are components of the normal, “healthy” microbiome [62]. Interestingly, diets with a reduced glycemic index seem not to impact inflammation markers or markers of cardiovascular risk in GDM, according to a recently published systematic review and meta-analysis [63].

Vitamin D supplementation seems to reduce the risk of GDM and displays a negative association with the values of the glycemia and of the homeostasis model of assessment for insulin resistance index (HOMA-IR). Moreover, vitamin D supplementation in GDM can alleviate glycemia, HOMA-IR, lipid, glutathione, and C-reactive protein levels [64]. In addition, vitamin D (if administered together with magnesium, zinc, and calcium) reduces inflammation and oxidative stress biomarkers, e.g., malondialdehyde C-reactive protein levels. Moreover, it increases antioxidant levels and decreases the birth weight and the number of macrosomic offspring born to mothers with GDM [65]. Moreover, GDM seems to be linked with high levels of serum folates and low levels of vitamin B12 [66]. Low serum concentrations of the latter have also been related to a high risk of preeclampsia [67].

## 3. Fetal Consequences of Maternal Gestational Diabetes Mellitus and Maternal Diet

Early diagnosis and treatment are mandatory in GDM as there are a number of severe complications that can occur in both the fetus and the mother. These include preeclampsia, premature birth, polyhydramnios, macrosomia, postnatal hypoglycemia, and jaundice, neonatal respiratory distress syndrome, and an increased risk of developing birth defects, the latter occurring in up to 10% of diabetic pregnancies [4,5,68]. Pregnancies with GDM are shorter and the proportion of cesarean sections is much higher than in normal pregnancies. In addition, among these pregnancies, macrosomia is 3 times higher, and shoulder dystocia 10 times more common [68,69].

Maternal hyperglycemia causes diabetic embryopathy, which represents multiple impairments in embryogenesis and diabetic fetopathy, translated as complications in fetal development [70]. Maternal hyperglycemia in the first trimester of pregnancy has an effect similar to that of ionizing radiation, hypoxia, alcohol, and high-risk drugs inhibiting the uptake of myoinositol which is indispensable in the stage of gastrulation and neurulation, resulting in congenital malformations, e.g., caudal regression, neural tube defects, atresia and digestive agenesis [71]. GDM with poor glycemic control causes fetal hyperglycemia (normally, the glycemia of the fetus is always 23–30 mg/dL below the maternal one), resulting in fetal hyperinsulinism and β-pancreatic cell hypertrophy [72]. These manifestations of fetal adaptation to the hyperglycemic environment cause organomegaly (especially cardiomegaly) and weight gain. Insulin excess causes the stimulation of fetal adipogenesis and leads to macrosomia [73]. Fetal macrosomia is defined as the fetal weight above the 90th percentile, i.e., over 4000 g; this value is calculated by a mathematical distribution of the birth weight of all newborns at 39 weeks of gestation [74].

The anabolic action of insulin is manifested at the fetal level by increasing the tissue production of amino acids and glucose while increasing the transplacental gradient for glucose and resulting in excessive intake [75]. Moreover, Naeye et al. (1965) reported that in postmortem specimens, the liver size of fetuses born to diabetic mothers was approximately 80% elevated versus healthy counterparts due to both cellular hyperplasia and hypertrophy and an elevated amount of hematopoietic tissue [76].

Fetal hyperinsulinism, in turn, causes stimulation of glycogen accumulation in the liver, increased lipid synthesis with the accumulation of subcutaneous adipose tissue, and disproportionate growth of insulin-sensitive tissues, namely, the liver tissue, muscle tissue including the myocardium, and the subcutaneous adipose tissue [77]. The fetus exposed to the hyperglycemic environment develops cardiomegaly with cellular changes, e.g., aggregation of ribosomes and vacuoles in the cytoplasm, structural changes in the myocardial tissue such as myoblast proliferation and increased rate of induction of apoptosis in myocardial cells, functional changes such as a high level of vascular endothelial growth factor (VEGF), and a very low amount of nitric oxide (NO) [78].

Fetal macrosomia in pregnancies complicated with GDM has a special feature, namely, that excess fat is deposited in the abdomen and scapular girdle, disproportionate to the cephalic extremity, leading to visceromegaly, with its onset after 24 weeks of gestation without any influence on the skeletal development of the fetus. Biometrically, macrosomic newborns from diabetic pregnant women with poor glycemic control are characterized by increased abdominal circumference, increased biacromial ratio, decreased skull–shoulder ratio, body fat deposition, and hypertrophic skin fold at the level of the upper extremities. Other features of macrosomic offspring born to pregnancies complicated with diabetes with poor glycemic control are the presence of cushingoid facies and the occurrence of skin jaundice [79].

The main risk factors for GDM and the main consequences of untreated GDM on the mother and offspring are represented in Figure 1.

Prospective randomized controlled studies to track the effects of maternal nutritional factors on the fetal liver are limited due to ethical implications. Thus, on human subjects, these effects can be followed only in retrospective studies. However, in the literature, there are data collected from animal studies that may show the possible effects of the maternal diet on fetal liver fat. A study on guinea pigs found that feeding the mother during pregnancy a Western diet is associated with a lower overall fetal fat level but with an increase in fetal liver fat (*p* < 0.02) [80]. In their study, Garcia-Contreras et al. demonstrated that maternal hydroxytyrosol supplementation alters the energy availability and content of fatty acids in the fetal tissues, diminishing the gross energy content of the fetal liver with an overall decreased amount of saturated fatty acids and an increased amount of polyunsaturated fatty acids [81]. Furthermore, Xue et al. demonstrated in a study conducted on sheep that maternal malnutrition is associated with changes in fetal metabolism such as increased oxidation processes and ketogenesis, increased triglyceride synthesis, decreased degradation of triglycerides and phospholipids, and decreased steroid synthesis [82].

## 4. The Size of Fetal Liver as a Predictive Parameter for the Evolution of GDM

Fetal growth and fetal liver development are influenced by the nutrient intakes of the fetus. The glucose tolerance of the mother and fetus and the insulin/insulin-like growth factor axis act as mediators of the relationship [83,84].

Fetal screening by ultrasound examination (2D, 3/4D) performed between 18 and 23 weeks of gestation is a noninvasive, effective, fast, and relatively inexpensive method of monitoring fetal development that can replace the OGTT as a diagnostic method for GDM [4,5]. The evaluation of fetal dimensions is performed by measuring some biometric indices. In addition to the standard ones evaluated in all routine OUS during pregnancy, such as fetal biparietal diameter, abdominal diameter, head diameter, or femur length, there are also some specific indices to the GDM-complicated pregnancy, such as fetal liver length (FLL) or volume (FLV), abdominal wall thickness, abdominal fat layer or Wharton’s gelatin thickness [4,5,37]. Of these, the most important is the FLL and the FLV because the liver of the fetus is directly influenced by the fetal blood glucose levels via excess glycogen deposition under the action of fetal insulin [4,37].

Given the direct relationship between GDM and various parameters related to fetal growth, particularly liver indices, which can be assessed by ultrasound, we wanted to evaluate the effectiveness of OUS as a method of diagnosis and monitoring complicated pregnancies with GDM.

Thus, we computed a search in PubMed/MEDLINE, Clarivate Analytics Web of Science, SCOPUS, and ScienceDirect for articles published up to 1 April 2021 that evaluated the relationship of fetal liver indices with GDM. For inclusion in this review, we selected the articles published in English, French, Italian, and Romanian (the languages spoken by the authors) with full texts that could be accessed and presented relevant information on OUS parameters useful in the evaluation of GDM. The exclusion criteria were (1) articles with full texts in another language than the aforementioned ones; (2) articles whose full-texts could not be accessed; (3) case reports, letters to the editor, reviews, or abstracts presented at various scientific conferences. The keywords and word combinations employed were “fetal liver”, “obstetrical ultrasound”, “gestational diabetes”, “pregnancy”, “gestational diabetes mellitus”, “midtrimester ultrasound”, “fetal liver length measurement”, “fetal growth”, “fetal liver blood flow”, “umbilical venous volume flow”, and the results are systematized in the following paragraphs and in Table 1.

## 5. Fetal Liver Length Measurements by Ultrasound—Any Value?

Roberts et al. (1994) analyzed a group of 104 pregnant women, diabetic or at risk to develop GDM (obesity: 24 subjects; type I diabetes: 26 subjects; type 2 diabetes: 54 subjects), each of them performing an OUS at 18, 28, and 36 weeks of gestation. The waist circumference, the length of the femur, and the FLL were notably elevated in pregnancy versus reference values (*p* < 0.001 for all time points). At each time point, the authors registered significantly higher FLL values (*p* < 0.001) [85]. Moreover, the mean excess size of the waist circumference and of the femur length remained roughly steady between 18 and 36 weeks of gestation, while the liver dimensions saw a significant rise (12.0% ↗ 16.7% ↗ 19.3% at 18, 24, and 36 weeks, respectively; *p* < 0.02). Despite this finding, there were no differences regarding fetal liver or other ultrasonographic measurements at any time point during gestation between type I and type 2 patients. All of these in utero measurements were maintained postpartum, with the liver of newborns from diabetic mothers weighing 179% of control values at delivery [85].

Mirghani et al. (2006) performed an elaborate OUS on a group of 123 pregnant women, 19 (15.4%) with GDM, and 104 (84.6%) healthy, based on the World Health Organization criteria. The ultrasound included, besides standard fetal biometry measurements and detailed anomaly scans, some specific evaluations of body composition, e.g., the Wharton’s jelly area, length of the right lobe of the liver, placental thickness, subcutaneous fat layer, and cardiac muscle. Of these, in women diagnosed with GDM, the FLL was the only measurement significantly elevated (*p* < 0.01) [69].

In the cross-sectional study of Boito et al. (2007), a group of 32 pregnant insulin-dependent diabetes mellitus (IDDM) females was compared to an equal control group of pregnant females with a normal status of health. As in the previous study, the authors examined in addition to common biometry measurements, such as fetal abdominal circumference or FLV, a series of particular indices, including liver volume/estimated fetal weight ratio, ultrasonically estimated fetal weight, umbilical venous volume flow per kilogram fetal weight or umbilical artery pulsatility index, using Doppler and B mode ultrasound [22]. The final statistical analysis demonstrated that both mean FLL and FLL/EFW ratios in the IDDM group were approximately 20% higher versus their healthy counterparts. Moreover, the IDMM subset registered statistically significant elevated values for the fetal liver volume (mL) and abdominal circumference (cm), ultrasonically EFW (g), and FLV/EFW ratio, and a statistically significant lower one was detected for the umbilical venous volume flow per kilogram fetal weight (mL/min/kg) [86].

Regarding the importance of the correct treatment, Dubé et al. (2011) analyzed a group of 27 pregnant women and divided them according to the OGTT results based on the guidelines of the Canadian Diabetes Association (CDA): 17 females had GDM (study group), and 10 females had normal glucose tolerance (NGT) (control group). The study found no differences in fetal weight or FLV in the GDM (managed according to the CDA guidelines) versus the NGT group. Thus, integrative management of GDM with a focus on stricter control of blood glucose levels may contribute to a normal weight at birth and a normal FLV. Moreover, the birth weight was associated with the FLV at 32 (ρ = 0.42, *p* = 0.03) and 36 (ρ = 0.61, *p* < 0.001) weeks, respectively [24].

Perovic et al. (2014) analyzed a group of 331 women at high risk of developing GDM, detecting a final GDM prevalence of 25.7%. There were significant differences only in terms of the BMI and the second parity (elevated in pregnant GDM females). GDM females, as opposed to healthy pregnant females, registered notably elevated fetal liver indices (*p* < 0.001). FLL correlated positively with the blood glucose levels during the OGTT (*p* < 0.001) (across all time points of the protocol) in GDM subjects, yet this correlation was not found in healthy patients (*p* > 0.05) [37]. Furthermore, FLL only correlated with the BMI (r = 0.586; *p* < 0.001) and not with the parity. FLL was significantly associated with the presence of GDM (odds ratio = 1.401; 95% confidence interval 1.308–1.501; *p* < 0.001; R^2^ = 0.597), and there was an independent association of FLL with the presence of GDM independently of BMI or parity. Using receiver operating characteristic curves, the authors decided a FLL cutoff value of 39 mm as a predictor of GDM (sensitivity = 71.76%, specificity = 97.56%, positive predictive value = 91.0%, negative predictive value = 90.9%) [37].

In a prospective study conducted by Ilhan et al. (2018), which enrolled 97 pregnant females, the study group was split based on the OGTT results: 64 healthy and 33 GDM-complicated pregnancies. The OUS included standard fetal biometric measurements and FLV. Despite no significant differences in terms of standard fetal biometric measurements and EFW (estimated fetal weight), FLV was notably elevated in the GDM versus *the* control group (*p* < 0.01) [4]. GDM females had significantly higher BMI, but there was no notable correlation observed between the BMI and the FLV in both the control (r = 0.169; *p* > 0.05) and GDM groups (r = 0.275; *p* > 0.05). On the other hand, although the birth weight was significantly elevated in the GDM group, the authors found a significant positive birth weight—FLV correlation in the GDM subjects (*p* < 0.05) [4].

In another prospective study, Showman et al. (2019) recruited 120 pregnant women aged 21–37 years at high risk for GDM. A routine OUS was performed at 23 weeks of gestation and followed up at 24 weeks by the OGTT, which detected an overall incidence of GDM of 19.2%. History of previous GDM and a first-degree family history of diabetes were the most important risk factors for GDM [5]. There was a strong association between the midtrimester FLL and the OGTT blood glucose values. Mean FLL values were significantly higher among GDM patients versus healthy pregnant females (37.2 (3.4) versus 33.1 (2.7); *p* < 0.001)), with the FLL in the GDM group being 1.6 times higher than in the non-GDM group (odds ratio = 1.6; 95% confidence interval 1.305–1.962) and having a high specificity (95.9%) and negative predictive value of 95.9% [5].

To assess the presence of dissimilarities between the ultrasonographic parameters of mothers with pre-GDM (PGD, i.e., the presence of diabetes before the debut of the pregnancy) or GDM, Gharib et al. (2019) divided 60 pregnant females (age 20–39) into two groups, based on the HbA1c and glycemia: healthy pregnant females as controls (n = 30) and pregnant females with PGD or GDM (n = 30) [65]. FLL was significantly elevated in PGD/GDM versus controls when measured at 28 weeks (48.9 ± 3.4 mm versus 41.7 ± 3.3 mm, *p* < 0.001) and again at 37 weeks of gestation (65.6 ± 4.8 mm versus 54.5 ± 3.4 mm, *p* < 0.001). Furthermore, when the authors compared the subjects within the case group, FLL was higher among those with PGD than those with GDM at 28 weeks (50.55 ± 2.35 mm versus 46.15 ± 2.1 mm, *p* = 0.01) and again at 37 weeks (66 ± 2.65 mm versus 59.69 ± 2.7 mm, *p* = 0.01). At 28 weeks, screening by FLL measurement exhibited a sensitivity of 100% and a specificity of 92% in predicting the occurrence of diabetes in pregnancy. In the end, there was a strong positive association of the FLL and the HbA1c (r = 0.83), AFI (r = 0.86), expected fetal birth weight (r = 0.82), abdominal circumference (r = 0.82), and neonatal birth weight (r = 0.80) [68]. This finding is similar to those of Szpinda et al. (2015), who found that FLV increased during pregnancy (6.57 cm^3^ ↗ 14.36 cm^3^ ↗ 20.77 cm^3^ at 18–21, 22–25, and 26–30 weeks of gestation, respectively), accelerated by approximately 20%/gestation week when compared with normal values [87].

Some animal and human studies showed that an increased hepatic flow from the umbilical vein results in elevated cell proliferation in different essential organs, such as the liver, heart, and kidneys. Fetal growth is regulated by the distribution of the nutrient-rich umbilical venous blood to the liver [88]. Moreover, higher hepatic umbilical venous flow is linked with offspring adiposity and larger fetal size. Based on these results, Lund et al. (2019) aimed in their prospective to study how glycemic control alters fetal hepatic blood flow in pregnancies complicated by PGD. Therefore, they analyzed fetal hemodynamic indices, e.g., mean left portal vein flow velocity or total venous supply using Doppler ultrasound on a group of 49 pregnant women with PGD (type 1 diabetes: 44 subjects; type 2 diabetes: 8 subjects). The mean left portal vein flow velocity and the total venous supply to the fetal liver in the PGD group were significantly higher than the reference, while the mean portal venous flow did not differ between the two groups. Overall, PGD pregnancies registered higher mean umbilical venous liver flows versus controls, particularly due to the high flows occurring <30 weeks of gestation [89].

In the prospective cohort study of Opheim et al. (2019), the impact of maternal nutritional conditions during fetal life over the long-term health of the newborn was explored. The consequences of regular maternal meals and of prepregnancy BMI on the fetal liver blood flow were assessed. Doppler OUS was employed in 137 healthy women at 18 and 20 weeks of gestation and several indices were evaluated, e.g., the umbilical vein flow (mL/min) or the mean left portal vein flow velocity [90]. Surprisingly, normal-weight subjects displayed a notable postprandial elevation in hepatic flow, whereas females with a high prepregnancy BMI displayed an opposite tendency. Moreover, a significantly higher umbilical venous liver blood flow in the normal-weight group regardless of fetal size was in contrast with the negative values reported in overweight subjects. The authors presumed that this effect is due to an “overnourished” environment found in the fetal liver of overweight mothers [90].

Overall, the analyzed research papers point out a possible connection between fetal liver indices and GDM; however, the base of evidence is too small to conclude that these parameters can serve as predictors of GDM in pregnancy. Consequently, further research is needed to clarify the relationship between fetal liver indices and GDM and, if our hypothesis is confirmed, to delineate cutoff values for the variables that could predict the development of GDM [5,24,37,68,69,85,86,87,88,89,90,91,92,93]. Only one study, conducted by Perovic et al. (2014), proposed an FLL of 39 mm as a predictor of GDM [37]. In the analyzed papers, the most investigated fetal liver indices were FLL (n = 5) and FLV (n = 5). Overall, the subjects with GDM had higher FLL and FLV versus comparators, with several studies revealing associations between FLL and (or) FLV and anthropometric (BMI or BW in particular) or carbohydrate metabolism indices (FPG or HbA1c) [5,24,37,68,69,85,86,87,88,89,90,91,92,93]. Interestingly, Boito et al. (2007) demonstrated that if glycemia is adequately controlled and if GDM is appropriately treated, the FLV is similar between GDM and NGT pregnancies [86]. However, as Opheim et al. (2019) revealed that overweight pregnant females have a decreased blood flow to the fetal liver, how nutrients reach the fetal liver to stimulate its growth requires further investigation. We may thus hypothesize that maternal–fetal nutrition and carbohydrate metabolism are not the only key players in the GDM–fetal liver equation. The findings of Opheim et al. (2019) might be limited by the small number of subjects (n = 21) diagnosed with overweight included in the analysis [90]. However, in a previous study, Haugen et al. (2004) revealed that the blood flow to the fetal liver was reduced, and less shunting of blood from the liver occurred in pregnant females who consumed an unbalanced diet or who were slim and had fewer stores of adipose tissue. The authors concluded that this “liver-sparing” phenomenon might be linked with the elevated cardiovascular risk of offspring born from these pregnancies [91]. Moreover, liver and cardiovascular disorders seem to share pathophysiological links also in adulthood, as previously reviewed by Jichitu et al. (2021) [92]. More recently, Opheim et al. (2020) revealed that the blood flow to the fetal liver was negatively associated with the maternal–fetal gradient of glucose and positively associated with the fraction of blood shunted via the ductus venosus in pregnant females with a normal BMI but not in those suffering from overweight. It seems that the fetal liver adapts to the energy supply of the mother [93]. However, Kamimae-Lanning et al. (2014) demonstrated that the hematopoiesis occurring in the fetal liver is dysfunctional if the pregnant mother is obese, or if she consumes a diet rich in fats, possibly explaining why the blood flow to the liver was reduced in overweight pregnant females in the study by Opheim et al. (2019), as a less functional liver probably requires fewer nutrients [90,94]. In addition, Ikenoue et al. (2021) pointed out that the blood flow to the liver is regulated by the concentrations of the placental corticotrophin-releasing hormone. In particular, the authors noted that at 30 weeks of pregnancy, a positive association exists between the concentrations of the hormone and the blood flow to the liver of the fetus (r = + 0.319; *p* = 0.004). Thus, the levels of corticotrophin-releasing hormone predicted approximately 10.5% of the blood flow to the liver of the fetus [95]. This can be noted as another limitation in the work of Opheim et al. (2019), as they did not assess the levels of corticotrophin-releasing hormone [90].

The main results of this subsection are summarized in Table 1.

**Table 1 ijms-22-07866-t001:** Fetal liver-related parameters evaluated by obstetric ultrasound.

Liver Ultrasound Timing (Weeks of Gestation)	No. of Subjects	Condition	Evaluated Parameters	Main Results	Reference
18, 28, 36	104	T1D, T2D, obesity	FL, WC, FLL, LS	FL↑, WC↑, FLL↑ versus reference values (*p* < 0.001) FLL↑ at all-time points during pregnancy (*p* < 0.001) Mean excess size of FL, WC: steady between 18–36 weeks ↑LS: 12.0% (18 weeks) → 16.7% (24 weeks) → 19.3% (36 weeks) (*p* < 0.02) T1D versus T2D: no differences at 18, 28, 36 weeks Postpartum: weight of newborns from diabetic mothers = 1.79 x controls	Roberts et al. (1994) [85]
21–24	123	GDM, healthy women	SFL, LRLL, CM, PT, WJA	LRLL↑ (*p* < 0.01) in GDM females FLV and maternal HbA1c were connected: liver volume is increased by 8.1% for each unit increase in HbA1c (95% CI 3.5–13.0%) and by 14% (95% CI 13.0–15.8%) per week of gestational age	Mirghani et al. (2006) [69]
18–36 (median 26)	64	IDDM, healthy women	FWC, FLV, FLV/EFWR, UEFW, UVV/kg FW	IDMM: ↑FLL, ↑FLV/EFWR = 1.20 x controls IDMM: ↑FWC, ↑FLV, ↑UEFW, ↑FLV/EFWR IDDM: ↓UVV/kg FW No differences in FLV at 32 and 36 weeks in NGT versus GDM if appropriate treatment	Boito et al. (2007) [86]
32, 36	27	GDM, NGT	FLV, FW	GDM versus NGT: no difference in FLV, FW FLV (32 weeks)-BW correlation (ρ = 0.42, *p* = 0.03) FLV (36 weeks)-BW correlation (ρ = 0.61, *p* < 0.001)	Dubé et al. (2011) [24]
23	331	GDM, healthy women	FLL	GDM: ↑BMI, ↑second parity, ↑ fetal liver measurements (*p* < 0.001) FLL-FPG positive correlation during OGTT (*p* < 0.001) FLL-BMI correlation (r = 0.586; *p* < 0.001) no FLL-parity correlation FLL-GDM association (OR = 1.401; 95% CI 1.308–1.501; *p* < 0.001; R2 = 0.597) independent of BMI/parity FLL = 39 mm, cutoff value for predicting GDM (sensitivity: 71.76%, specificity: 97.56%, positive predictive value: 91.0%, negative predictive value: 90.9%)	Perovic et al. (2014) [37]
24–28	97	GDM, healthy women	FLV, EFW	no differences in standard fetal biometric measurements, EFW GDM: ↑FLV (*p* < 0.01), ↑BMI, ↑BW no FLV-BMI correlation BW-FLV positive correlation (*p* < 0.05)	g et al. (2018) [4]
24	120	GDM, healthy women	FLL	midtrimester connection of FLL and FPG (OGTT)GDM: ↑FLL [37.2 (3.4)] versus controls [33.1 (2.7)], *p* < 0.001 FLL (GDM) = 1.6 x controls (OR 1.6; 95% CI 1.305–1.962), specificity 95.9%, negative predictive value 95.9%	Showman et al. (2019) [5]
28, 37	60	PGM, GDM, healthy women	FLL	PGM, GDM vs. controls: ↑FLL (28 weeks), 48.9 ± 3.4 mm vs. 41.7 ± 3.3 mm, *p* < 0.001 PGM, GDM vs. controls: ↑FLL (37 weeks), 65.6 ± 4.8 mm vs. 54.5 ± 3.4 mm, *p* < 0.001 PGD vs. GDM: ↑FLL (28 weeks), 50.55 ± 2.35 mm vs. 46.15 ± 2.1 mm, *p* = 0.01 PGD vs. GDM: ↑FLL (37 weeks), 66 ± 2.65 mm vs. 59.69 ± 2.7 mm, *p* = 0.01 FLL correlated with WC (r = 0.82), AFI (r = 0.86), HbA1c levels (r = 0.83), EFBW (r = 0.82), BW (r = 0.80)	Gharib et al. (2019) [68]
18–21, 22–25, 26–30	69	Healthy human fetuses	FLV	↑FLV 6.57 cm^3^ (18–21 weeks) → 14.36 cm^3^ (22–25) → 20.77 cm^3^ (26–30 weeks) ↑FLV by 20%/week of gestation vs. normal	Szpinda et al. (2015) [87]
24–36	49	PGM: T1D, T2D	LPVFV, TVSPFL, UVLF	↑LPVFV, ↑TVSPFL vs. reference no difference in PVF ↑UVLF in GDM vs. reference mean	Lund et al. (2019) [89]
18, 20	137	Healthy women	LF, UVLF	postprandial ↑ liver flow in NW postprandial ↓ liver flow if ↑BMI prepregnancy ↑UVLF in NW regardless of fetal size ↓UVLF in the overweight	Opheim et al. (2019) [90]

Abbreviations: GDM, gestational diabetes. FL, femur length. WC, waist circumference. FLL, fetal liver length. T1D, type 1 diabetes. 270 T2D, type 2 diabetes. GDM, gestational diabetes. SFL, subcutaneous fat layer. LRLL, length of the right lobe of the liver. CM, cardiac 271 muscle. PT, placental thickness. WJA, Wharton’s jelly area. IDDM, insulin-dependent diabetes mellitus. FWC, fetal WC. FLV, fetal 272 liver volume. FLV/EFWR, FLV/estimated fetal weight ratio. UEFW, ultrasonically estimated fetal weight. UVV/kg FW, umbilical 273 venous volume flow per kilogram fetal weight. UAPI, umbilical artery pulsatility index. NGT, normal glucose tolerance. FPG, fasting 274 plasma glucose. BW, birth weight, OR odds ratio. EFW, estimated FW. PGD, pre-GDM. vs., versus. EFBW, estimated fetal BW. 275 LPVFV, left portal vein flow velocity. TVSPFL, total venous supply to the fetal liver. PVF, portal venous flow. UVLF, umbilical venous 276 liver flow. UVF, umbilical vein flow. NW, normal weight, LF, liver flow. ↑ increased. ↓ decreased. → to/at.

Most authors who compared fetal liver measurements obtained by OUS found significant differences between the values measured in women with GDM versus healthy counterparts (*p* < 0.001) in terms of FLV: 41.46 ± 6.56 cm^3^ versus 33.67 ± 6.42 cm^3^ [4]. In Showman et al. (2019)’s research, FLL was a useful parameter in the early detection of GDM due to its elevated specificity of 95.9% and its role as a negative predictor in 95.9% of cases. The results showed that the mean FLL value in the GDM group was 1.6 times the value in the non-GDM group (37.2 mm vs. 33.1 mm; *p* < 0.001) [5]. Similar results were obtained by Gharib et al. (2019), particularly 48.9 ± 3.4 mm (40.4–55) in the GDM group vs. 41.7 ± 3.3 mm (34.5–49.2) in the control group, *p* < 0.001 and Mirghan et al. (2006), particularly 36 (32–37) mm vs. 31 (30–33 mm, *p* < 0.001 [68,69]. At the same time, even though Perovic et al. (2014) established a cutoff value for FLL of 39 mm for GDM prediction, with a sensitivity of 71.76%, specificity of 97.56%, a positive predictive value of 91.0%, and a negative predictive value of 90.9%, more research is still needed in this field to be able to draw a clear conclusion and, in the absence of well-established international guidelines, FLL will remain at this point only a complementary index and not a standard one for the early diagnosis of GDM [37].

## 6. The Value of Nutrition Therapy in GDM

Currently, nutritional and lifestyle interventions have been recognized as the cornerstone of therapy for females diagnosed early with GDM. These approaches have emerged as attractive strategies with benefits that extend beyond pregnancy, being particularly helpful in decreasing the risk of CVD or T2DM [96,97,98]. It is estimated that 70–85% of cases can be controlled with such interventions alone [99]. These strategies are based on caloric restriction, the control of carbohydrate intake, and physical activity within tolerability limits. Some of the eight globally recognized diets that help pregnant women lose weight are the MedDiet or the DASH diet [96,98].

The caloric restriction remains a foundational strategy in preventing ponderal gain, controlling glycemia values, and preventing macrosomia in the offspring born to GDM mothers [96]. A strict dietary approach (based on an amount of 1500 daily, i.e., 50% reduction) has led to ketonuria and ketonemia, but a more moderate one has been more successful, managing to control weight gain and glucose levels without increasing ketonemia [96]. One study showed that decreasing the BMI by >2 points results in a subsequent decrease of the GDM risk by 74%, whereas an elevation of the BMI nearly doubles the risk of GDM [97].

Physical activity has shown multiple benefits, such as improving blood glucose control, reducing weight, insulin resistance, and cardiovascular risk. Thus, regular physical exercise might play an important role in GDM prevention [96,97,100,101]. Some studies showed a rapid effect of reducing glucose levels by 23 mg/dL at 30 min and a 69% reduced risk of GDM if sustained physical activity was performed [96,97]. Usually, if the target blood glucose levels are not reached within 1–2 weeks, pharmacotherapy should be initiated [99,102]. Historically, when that happened, the sole alternative was insulin because oral antidiabetic medications were contraindicated during pregnancy due to the possible risks of teratogenicity and life-threatening neonatal hypoglycemia [102]. Today, the most prescribed oral antidiabetics during pregnancy are metformin and glyburide, which, although not approved, are not banned by the United States Food and Drug Administration (FDA) and are recommended by a few key organizations, including the American Congress of Obstetricians and Gynecologists (ACOG), the Society of Maternal–Fetal Medicine (SMFM), or the American Diabetes Association (ADA) [15,16,17].

## 7. Conclusions

Given all the above, we conclude that an early diagnosis of GDM is crucial due to its potential complications, i.e., preeclampsia, birth defects, and possible development of CVD and T2DM later in the life of the newborn. Screening all pregnancies with an OGTT may not always be feasible due to several drawbacks. Since a midtrimester OUS is already a standard, future studies should investigate its feasibility and utility in the prediction, early diagnosis, and follow-up of GDM and, additionally, in estimating the birth weight prenatally. Measuring different fetal liver indices is an easy technique and could emerge as a reliable method to assess GDM pregnancies. Further research should clarify whether common measurement parameters, i.e., FLL and FLV, could be strong predictors of GDM and to which extent they positively relate to maternal HbA1c levels. In addition, other indirect indicators, such as fetal liver blood flow, have been shown to be strongly connected to the glycemia of the GDM female in the first trimester of pregnancy. Finally, these studies highlighted the crucial role of a proper multidisciplinary approach to GDM treatment during pregnancy and maternal nutritional status, as the enhanced growth of the fetal liver can be modulated by controlling the mother’s glycemia even in the late stages of pregnancy. Soon, medical nutrition therapy should also be integrated into the management of pregnancies at risk for GDM.

## Figures and Tables

**Figure 1 ijms-22-07866-f001:**
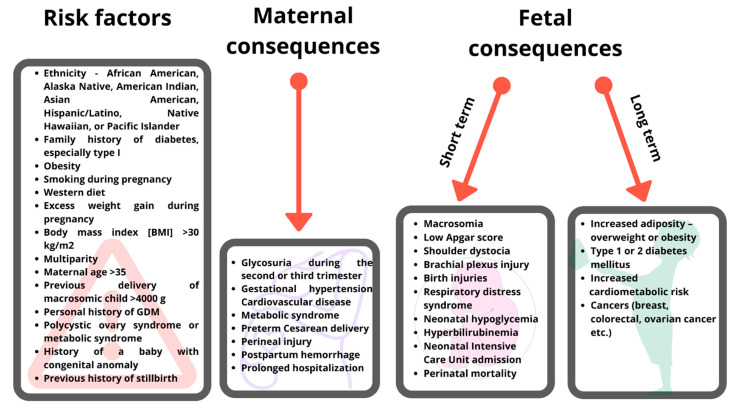
Consequences of gestational diabetes mellitus (GDM) on the mother, fetus, and offspring.

## Data Availability

Not applicable.

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
