# Peer review of "A Glimpse at the Size of the Fetal Liver—Is It Connected with the Evolution of Gestational Diabetes?"

_ijms, 2021, doi:10.3390/ijms22157866_

Round 1

Reviewer 1 Report

General comments: This is interesting and well written manuscript with formulated concrete aims, with the literature search described in details, including search terms and inclusion criteria. Furthermore, relevant endpoint data are presented appropriately. However, some minor revision is needed in order to further increase the quality of the manuscript.

Specific comments:

The 2nd pharagraph on page 3 (lines 110-120) is starting with the sentence "The standard diagnostic method for GDM is the oral glucose tolerance test (OGTT) 110 (75 g)…" and it describes diagnostic methods for GDM (OGTT, fasting blood glucose, and glycated hemoglobin). However, modern diagnostic approach in detecting GDM uses ultrasonography as complementary tool in detection of GDM (references:

- Perović M, Garalejić E, Gojnić M, et al. Sensitivity and specificity of ultrasonography as a screening tool for gestational diabetes mellitus. J Matern Fetal Neonatal Med. 2012 Aug;25(8):1348-53.

- Jin D, Rich-Edwards JW, Chen C, et al. Gestational Diabetes Mellitus: Predictive Value of Fetal Growth Measurements by Ultrasonography at 22-24 Weeks: A Retrospective Cohort Study of Medical Records. Nutrients. 2020 Nov 27;12(12):3645.

- Gojnic M, Stefanovic T, Perovic M, et al. Prediction of fetal macrosomia with ultrasound parameters and maternal glycemic controls in gestational diabetes mellitus. Clin Exp Obstet Gynecol. 2012;39(4):512-5.).

I suggest you to add this into your manuscript, especially because one of the most important tools in ultrasound detection of GDM is evaluation of the Fetal Liver (references:

- İlhan G, Gültekin H, Kubat A, et al. Preliminary evaluation of foetal liver volume by three-dimensional ultrasound in women with gestational diabetes mellitus. J Obstet Gynaecol. 2018 Oct;38(7):922-926.

- Perovic M, Gojnic M, Arsic B, et al. Relationship between mid-trimester ultrasound fetal liver length measurements and gestational diabetes mellitus. J Diabetes. 2015 Jul;7(4):497-505.)

The 3rd paragraph, page 5, lines 231-241 beginning with "The anabolic action of insulin is manifested at the fetal level by increasing the tissue…" should include data regarding reasons (other then glycogen accumulation in the liver)  for enlargement of fetal liver. For example, Naeye reported that, in post-mortem specimens, the liver size of fetuses from diabetic mothers was increased by approximately 80% compared with normal controls because of both cellular hyperplasia and hypertrophy and an increased amount of hematopoietic tissue (reference: Naeye RL. Infants of diabetic mothers: A quantitative, morphologic study. Pediatrics. 1965; 35: 980–8.).

Author Response

Dear Academic Editor,

Dear Peer-Reviewers,

We are very thankful to you and to the peer-reviewers for the pertinent notes; we have carefully read the comments and have revised/completed the manuscript accordingly. Our responses are given in a point-by-point manner below. All the changes to the manuscript are highlighted in yellow. We hope that, in this new form, the manuscript will be suitable for publication in the International Journal of Molecular Sciences (IJMS).

Reviewer 1

  • General comments: This is interesting and well written manuscript with formulated concrete aims, with the literature search described in details, including search terms and inclusion criteria. Furthermore, relevant endpoint data are presented appropriately. However, some minor revision is needed in order to further increase the quality of the manuscript.

Response: Thank you for your valuable suggestions and for your appreciation of our manuscript.

Specific comments:

  • The 2nd pharagraph on page 3 (lines 110-120) is starting with the sentence "The standard diagnostic method for GDM is the oral glucose tolerance test (OGTT) 110 (75 g)…" and it describes diagnostic methods for GDM (OGTT, fasting blood glucose, and glycated hemoglobin). However, modern diagnostic approach in detecting GDM uses ultrasonography as complementary tool in detection of GDM (references):

- Perović M, Garalejić E, Gojnić M, et al. Sensitivity and specificity of ultrasonography as a screening tool for gestational diabetes mellitus. J Matern Fetal Neonatal Med. 2012 Aug;25(8):1348-53.

- Jin D, Rich-Edwards JW, Chen C, et al. Gestational Diabetes Mellitus: Predictive Value of Fetal Growth Measurements by Ultrasonography at 22-24 Weeks: A Retrospective Cohort Study of Medical Records. Nutrients. 2020 Nov 27;12(12):3645.

- Gojnic M, Stefanovic T, Perovic M, et al. Prediction of fetal macrosomia with ultrasound parameters and maternal glycemic controls in gestational diabetes mellitus. Clin Exp Obstet Gynecol. 2012;39(4):512-5.).

Response: Thank you for your valuable comments and for suggesting these excellent references. We have included them in the following paragraph – see references no. 38, 39, and 40.

Although the above tests have an undeniable diagnostic value, new non-invasive diagnostic methods, e.g., ultrasonography, have emerged as useful tools in identifying patients at risk of developing GDM. Thus, Perovic et al. (2012) highlighted the advantages of an ultrasound GDM screening score as a predictor of GDM development by screening pregnant females who were at least in the week 24 of gestation and who harbored risk factors for GDM. The ultrasound GDM screening score proposed by the aforementioned researchers exhibited a specificity and a sensitivity of over 89% and took into consideration several relevant parameters, i.e., subcutaneous fetal adipose tissue, the immature appearance of the placenta, and placental thickness [38]. Moreover, another study by Gojnic et al. (2012) revealed that subcutaneous fetal adipose tissue exhibited the best specificity and sensitivity in predicting GDM if the evaluation was performed in week 32 of gestation [39]. Furthermore, prior to week 24 of gestation, other ultrasound parameters, i.e., head circumference below the 10th percentile and femur length below the 10th percentile have been linked with a 13-17% elevated risk of GDM [40].

  • I suggest you to add this into your manuscript, especially because one of the most important tools in ultrasound detection of GDM is evaluation of the Fetal Liver (references:

- İlhan G, Gültekin H, Kubat A, et al. Preliminary evaluation of foetal liver volume by three-dimensional ultrasound in women with gestational diabetes mellitus. J Obstet Gynaecol. 2018 Oct;38(7):922-926.

- Perovic M, Gojnic M, Arsic B, et al. Relationship between mid-trimester ultrasound fetal liver length measurements and gestational diabetes mellitus. J Diabetes. 2015 Jul;7(4):497-505.)

Response: Thank you for your valuable comments and for suggesting these excellent references. We have included them in our manuscript. Please see references no. 4 and 37.

  • The 3rd paragraph, page 5, lines 231-241 beginning with "The anabolic action of insulin is manifested at the fetal level by increasing the tissue…" should include data regarding reasons (other then glycogen accumulation in the liver) for enlargement of fetal liver. For example, Naeye reported that, in post-mortem specimens, the liver size of fetuses from diabetic mothers was increased by approximately 80% compared with normal controls because of both cellular hyperplasia and hypertrophy and an increased amount of hematopoietic tissue (reference: Naeye RL. Infants of diabetic mothers: A quantitative, morphologic study. Pediatrics. 1965; 35: 980–8.).

Response: Thank you for your valuable comments and for suggesting this excellent reference. We have included it in the following paragraph – see reference no. 76.

Moreover, Naeye et al. (1965) reported that, in post-mortem specimens, the liver size of fetuses born to diabetic mothers was approximately 80% elevated versus healthy counterparts due to both cellular hyperplasia and hypertrophy and an elevated amount of hematopoietic tissue [76].

Reviewer 2 Report

In the review by Cozma et al., the Authors discussed on gestational diabetes mellitus (GDM), focusing their attention on the relation between GDM and fetal nutritional status. They discussed as fetal liver growth could be a predictor factor for GDM evolution. The manuscript is well-written, the topic interesting  and references up-to-date. The words in Figure 1 should be more sharp.

Author Response

Dear Academic Editor,

Dear Peer-Reviewers,

We are very thankful to you and to the peer-reviewers for the pertinent notes; we have carefully read the comments and have revised/completed the manuscript accordingly. Our responses are given in a point-by-point manner below. All the changes to the manuscript are highlighted in yellow. We hope that, in this new form, the manuscript will be suitable for publication in the International Journal of Molecular Sciences (IJMS).

Reviewer 2

This is the first review correlating liquid biopsy with myeloproliferative syndromes. It is well done and provides relevant information. I just wanted to make a few minor comments

Response: We would like to thank you for your valuable comments which helped us improve the manuscript. All suggestions were taken into consideration and appropriate information, as well as required corrections, were provided. New/corrected parts are highlighted in yellow to facilitate the assessment of changes. We did our best to fulfill the expectations and we hope that you will be satisfied with our corrections.

All in all, we thank you for your positive comments and appreciation regarding our manuscript.

In the review by Cozma et al., the Authors discussed on gestational diabetes mellitus (GDM), focusing their attention on the relation between GDM and fetal nutritional status. They discussed as fetal liver growth could be a predictor factor for GDM evolution. The manuscript is well-written, the topic interesting and references up-to-date. The words in Figure 1 should be more sharp.

Response: Thank you for your valuable comment. We have revised Figure 1 to ensure that the words are sharper.
